# Implementation of an Advance Care Planning Inventory and Its Possible Effect on Quality of Dying: A Nationwide Cross-Sectional Study in Group Homes for Persons with Dementia in Japan

**DOI:** 10.3390/healthcare10010062

**Published:** 2021-12-29

**Authors:** Seira Takada, Yasuko Ogata, Yoshie Yumoto, Masaomi Ikeda

**Affiliations:** 1Department of Gerontological Nursing and Healthcare Systems Management, Graduate School of Health Care Sciences, Tokyo Medical and Dental University (TMDU), Tokyo 113-8510, Japan; yogata.gh@tmd.ac.jp (Y.O.); yumoto.gh@tmd.ac.jp (Y.Y.); 2Department of Gerontological Nursing, Faculty of Nursing, Toho University, Tokyo 143-0015, Japan; 3Department of Oral Prosthetic Engineering, Graduate School of Medical and Dental Sciences, Tokyo Medical and Dental University (TMDU), Tokyo 113-8510, Japan; ikeda.csoe@tmd.ac.jp

**Keywords:** dementia, nursing care, advance care planning, group homes, long-term care

## Abstract

This study aimed to develop an inventory for advance care planning implementation for persons with dementia in group homes and to examine the association between inventory implementation and residents’ quality of dying. A nationwide cross-sectional study was conducted via questionnaires mailed from 2000 group homes in Japan, selected through stratified random sampling. Participants were managers and care planners who had provided end-of-life care for recently deceased residents. The newly developed inventory was used to assess advance care planning implementation for persons with dementia, and the Quality of Dying in Long-term Care Scale was used to evaluate quality of dying. The valid response rate was 28.5% (*n* = 569). The factor structure of the newly developed Advance Care Planning Practice Inventory and the association between its implementation and quality of dying were verified using factor analysis and internal consistency, and logistic regression, respectively. The composite score and the factor score of the newly developed inventory were significantly associated with quality of dying (*p* < 0.05). The implementation of advance care planning improves the quality of dying. These findings can be used in development of educational programs, as well as research on advance care planning for care providers.

## 1. Introduction

The number of persons with dementia (PWD) is expected to increase worldwide as the population ages, and Japan is no exception [1,2]. As dementia symptoms progress, PWD find it difficult to express their wishes and live independently; some move into nursing and aged care facilities. Group homes for PWD are one such type of residence—a community-based service covered by long-term care insurance that started in Japan in 2000 [3]. The facility has a typical capacity of five to nine residents per unit, with a 3:1 resident-to-staff ratio during the day. Care is provided primarily by caregivers, and group homes should not be staffed with medical staff [4]. An additional end-of-life care bonus is allotted to group homes for PWD to strengthen end-of-life care [5], and more than half of the group homes for PWD provide end-of-life care in collaboration with visiting nurses and commissioned doctors. In addition, more than 60% of residents’ family members wish for the residents to spend their final days in the group home, with which they are familiar [6]; this is expected to further promote end-of-life care. To improve the quality of end-of-life care in these facilities, the practice of advance care planning (ACP) is comparably important to the home facilities themselves [7].

ACP involves facilitating adults at any age or health stage to comprehend and share their values, goals, and preferences pertaining to future medical care [8]. Its effectiveness has been well-researched, and its implementation is recommended to improve the quality of end-of-life care [9,10]. ACP is preferred, in part, because it ensures that patients die where they desire; this is desirable since it not only accounts for the needs and wants of the dying patient but also reduces the psychological burden on family and staff [11,12,13]. However, Dixon et al. [9] suggest that ACP may be underestimated in intervention studies because of a “black box” that does not clearly show how ACP is implemented; this is an issue that needs to be resolved in quantitative studies as well. To assess the effectiveness of ACP, it is necessary to clarify and evaluate its contents and to examine its relationship to the quality of end-of-life care.

Therefore, we developed an inventory to assess the implementation of ACP for PWD in group homes in order to clarify the structure of ACP and to examine the association between implementation of the inventory and quality of dying.

## 2. Materials and Methods

### 2.1. Participants and Procedure

In this nationwide cross-sectional study, using a publicly available database [14], 2000 facilities were randomly selected through stratified sampling by prefecture from approximately 6000 group homes for PWD that provide end-of-life care. Since the response rate tends to be low in national surveys of group homes in Japan, this number was deemed appropriate to ensure a sufficient sample size for factor analysis (4–10 times the number of variables) [15].

Data collection took place from August to September 2020. There were two types of questionnaires: one for managers and one for care planners for residents who had recently died in the facility. The questionnaires, along with return envelopes, was mailed to target facilities, and the respective managers distributed them to care planners. To increase the response rate, research assistants explained the survey outline to the managers over the phone, and two bi-weekly reminders were sent to the facilities after questionnaire distribution. The researchers conducted an online briefing for each research assistant, provided them with a manual to follow, and explained the instructions. The questionnaires were merged based on a pre-assigned identification number (ID). To ensure anonymization, the IDs were provided by a third-party agency, and the researchers were not informed of the assigned IDs. We merged the questionnaires based on IDs and the valid responses received that met the following inclusion criteria: (a) responses were received from both the manager and care planner and (b) none of the variables used in the analysis was missing.

### 2.2. Measures

#### 2.2.1. Demographic Characteristics

The questionnaire included items on participants’ age, sex, years of experience at the current facility, and job type. The questionnaire for managers included items about the facility’s characteristics (e.g., organization, year of opening, resident capacity, and average level of long-term care required); for care planners, it included items about the deceased resident’s characteristics (e.g., age, sex, length of stay, level of independence, quality of dying, and ACP Practice Inventory).

#### 2.2.2. ACP Practice Inventory (ACP-PI)

This inventory indicates the ACP tasks for PWD living in group homes. The authors conducted a scoping review in 2017 following a guideline [16] to identify ACP components for older adults with dementia living in senior living facilities. We developed a list of 38 items after the review. We combined several factors to constitute the ACP-PI from the Japanese ACP guidelines [17,18], such as “Discuss end-of-life intentions with the residents themselves” and “Support decision making by considering the cognitive functions of residents”. Items included “We talked with Mr./Ms. A about where they would like to spend the final stages of life”, “We talked with Mr./Ms. A about what kind of medical treatment/care they wished for at the final stage of life”, “We considered the topics of discussion regarding the final stage of Mr./Ms. A’s life depending on their cognitive functioning”, and others. We interviewed seven staff members working in group homes to determine the feasibility of implementing the items. Additionally, an expert panel of five experts in dementia and institutional care (e.g., Certified Nurse Specialists in gerontology and senior-citizen facility administrators) was convened to review the feasibility, necessity, and validity of each item and verify the content validity. The experts rated each item on a scale of 1–9 for validity; average score ≥7 was considered appropriate [19], and 33 items were classified as appropriate. Five items (score range: 5.8–6.6) were less than 7 (range 5.8–6.6) and were grouped as “uncertain” (score range: 4–6). As these five items were classified as “uncertain” and not “inappropriate” (less than 3), they were used directly in the next survey. Based on their recommendations, all items were included, and one more item was added because it was important in decision making to make sure that the PWD understood what was being discussed. A pilot survey was conducted in 2019 in nursing homes with characteristics similar to those of group homes. However, since the response rate for this pilot survey was very low (13%), we decided that it would be ineffective to eliminate or modify items based on the results because the low response rate could lead to the accidental deletion of some important items. Thus, the 39 items used in the pilot test were directly used in this study as draft items for the ACP-PI. Each item is rated on a 4-point Likert scale (1: “Did not implement,” 2: “More or less did not implement,” 3: “More or less implemented,” and 4: “Implemented”).

#### 2.2.3. Quality of Dying

The Quality of Dying in Long-term Care (QOD-LTC) Scale, a reliable and valid instrument, comprises three factors (personhood, preparatory tasks, and closure) across 11 items [20]. It uses a 5-point Likert scale, with higher scores indicating higher quality of end-of-life care. For the sub-factors, the average score for items consisting of the factor is calculated. The overall score is then calculated as the average of the three factor scores (range: 1–5 for the sub-scale and total scores). The scale was translated into Japanese with the developer’s permission. Cronbach’s alpha in this study was 0.80 for all QOD-LTC items and 0.75, 0.55, and 0.63 for the three factors—personhood, preparatory tasks, and closure—respectively. Since the QOD-LTC scores were non-normally distributed, this study used a dichotomous distribution according to the median, where scores ≤3 were categorized as “low group”, and scores >3 were categorized as “high group”.

#### 2.2.4. Covariates

To examine the association between ACP-PI and QOD-LTC, resident- and facility-related variables predicted to be associated based on existing studies and empirical data were used as covariates. Resident-related variables included the resident’s sex, age at death, years of residence, and functioning before death [10]. Facility-related variables included the acquisition of the end-of-life care bonus and number of full-time nurses. Life functioning before the death of PWD was assessed by the staff based on functional assessment staging, which represents the stage of functional impairment in activities of daily living due to dementia [21]. In this study, the cut-off score for severe dementia—little to no speech and requiring full assistance—was 7, and scores ≤6 indicated less than moderately severe dementia [22].

The facilities were required to meet the following requirements to receive the end-of-life bonus: 24-h nurse availability, explaining a policy for end-of-life care to the patient and/or family, providing care with the consent of the patient or their family, and having an end-of-life care system through the Plan-Do-Check-Action cycle [23].

#### 2.2.5. Japanese Version of the Frommelt Attitudes toward Care of the Dying Scale, Short Version (FATCOD-B-J-S)

This instrument measures staff attitude toward dying patients (six items and two factors) [24,25]. Items 2 and 3 are reversed items, so the score is set to “subtract the score from 6.” The score for each sub-factor is the sum of items [range: 3–15], and the total score is the sum of sub-factors [range: 6–30]. In this study, only care planners’ responses to one of the sub-scales, “positive attitudes toward caring for dying persons,” were used.

### 2.3. Analysis

After computing the descriptive statistics, the ACP-PI’s validity and reliability were examined. Logistic regression analysis was then performed.

#### 2.3.1. Validity and Reliability of the ACP-PI

An exploratory factor analysis (EFA) and a confirmatory factor analysis (CFA) were performed after confirming item distribution in order to identify the structure of the items in the ACP-PI. Both EFA and CFA are methods of measuring constructive validity. The first step is to conduct EFA and establish the factors of the inventory; then, CFA is used to validate the established factors. Data from the same group home were randomly divided into two datasets: 40% for EFA (*N* = 228) and 60% for CFA (*N* = 341). EFA was performed using principal axis factoring and promax rotation. Before the EFA, Bartlett’s and Kaiser–Meyer–Olkin tests were performed. The number of factors was determined by eigenvalues >1, scree plots, and factor interpretability. Items with factor loadings <0.45 were deleted. We performed CFA using weighted least square mean and variance adjusted (WLSMV) because these are used for ordinal data. The sample size after splitting satisfied the sample size required for WLSMV (*N* > 300) [26]. The criteria for the model’s goodness of fit were as follows: comparative fit index (CFI), the Tucker–Lewis index (TLI) >0.95, root–mean–square error of approximation (RMSEA) <0.06, and standardized root–mean–square residual (SRMR) <0.08 [27,28].

To test the criterion-related validity of the ACP-PI items, Spearman’s correlation analysis was performed using sub-factor scores of the FATCOD-B-J-S and QOD-LTC as external criteria. The QOD-LTC sub-factor, “preparatory tasks”, includes items such as “documenting treatment wishes” and “nominating a surrogate decision maker”. The “advance directive” included as a component of the ACP is similar to this documentation and nominating and was expected to have a moderately positive correlation with the “preparatory tasks” of the QOD-LTC. The FATCOD-B-J-S sub-scale, “positive attitudes toward caring for dying persons”, includes items such as “I feel uncomfortable talking about death with dying patients”; it was expected to have a small-to-moderate positive correlation with the ACP-PI. Cronbach’s alpha coefficient was calculated to confirm the internal consistency of the ACP-PI [29].

#### 2.3.2. Logistic Regression Analysis

Logistic regression analysis was performed using the simultaneous entry method, with the QOD-LTC as the dependent variable and scores on ACP-PI, facility, and resident demographic variables as the independent variables. To avoid multicollinearity, Spearman’s correlation analysis was performed, and variables with a correlation coefficient ≥0.7 were excluded [30]. In the analysis, the composite score and the score of each factor of the ACP-PI were used in the following models: for Model 1, only the composite score of the ACP-PI was entered; for Models 2–4, each factor of the ACP-PI was entered separately; for Model 5, all factors of the ACP-PI were included simultaneously. Other variables were used in all models. The significance level was set at *p* < 0.05. SPSS Statistics Ver. 27 for Windows and Mplus ver. 8.5 for Windows were used for analysis.

### 2.4. Ethical Considerations

The Medical Research Ethics Committee of Tokyo Medical and Dental University approved this study (approval number: M2019-064). Participants were informed in writing to respond freely to the questionnaire. The responses were anonymous, and those who consented to participate were included in the analysis.

## 3. Results

Figure 1 presents the flow chart for participants included in the study and final analysis. The valid response rate was 28.5% for both managers (*n* = 569) and care planners (*n* = 569); it was calculated by dividing the number of responses by the number of distributions (*n* = 2000) after excluding missing or incomplete responses. Using the pre-assigned IDs, we merged the responses of managers and care planners at the same facility into one set, resulting in a rate of 28.5% (*n* = 569) valid responses for the set.

Table 1 presents the characteristics of the facilities, managers and care planners, and Table 2 presents those of the recently deceased residents.

### 3.1. Validity and Reliability of the ACP-PI

Items with skewed distributions were eliminated (Table 3), resulting in 22 items. In the EFA, four items were excluded due to low factor loadings and commonality, resulting in 18 items and 3 factors: “provision of information and conversation with the resident to encourage them to express their end-of-life wishes”, “preparations in case the resident becomes unable to express their own end-of-life wishes”, and “devising ways to encourage the resident to express their wishes with consideration for their dementia”. Table 4 shows the results of the EFA and the Cronbach’s alpha coefficients.

After establishing the factor structure through EFA, CFA was conducted to confirm the factor structure. The CFA results for the initial model were: RMSEA = 0.101 (90% confidence interval (CI) 0.093−0.109), CFI = 0.980, TLI = 0.977, and SRMR = 0.041. The paths of the error correlations based on the modification indices were added, and the results were: RMSEA = 0.069 (90% CI 0.060−0.078), CFI = 0.991, TLI = 0.989, and SRMR = 0.033. Since the number of items differed among the factors, each factor’s score was calculated as the sum of the item scores for each factor divided by the number of items in that factor. The composite score for items of the ACP-PI was calculated by summing each factor’s scores and dividing by the number of factors. The composite scores and the scores for each factor are also shown in Table 4. The final version of the ACP-PI is shown in the Table 5.

In the Spearman’s correlation analysis, the correlation coefficient between the composite score and “preparatory tasks” (sub-scale of the QOD-LTC) was r = 0.34, *p* < 0.001, and that of “positive attitudes toward caring for dying persons” (sub-scale of the FATCOD-B-J-S) was r = 0.14, *p* < 0.001.

### 3.2. Association between ACP-PI and QOD-LTC

The results of Spearman’s correlation analysis among the independent variables did not show any correlation coefficient ≥0.7. The results of the logistic regression analysis showed a significant relationship between the QOD-LTC and the composite score, as well as each factor score of the ACP-PI, age at death, and additional end-of-life care bonus (Table 6). In model 5, where all three factors were entered simultaneously, only “provision of information and conversation with the resident to encourage them to express their end-of-life wish” was significantly associated with the QOD-LTC.

## 4. Discussion

This is the first study to develop an inventory for ACP-PI implementation in order to identify the factors and structure of ACP for PWD in group homes and to examine the association between inventory implementation and the residents’ QOD-LTC.

### 4.1. Structure of the ACP-PI for Group Homes for PWD

In developing the ACP-PI, the validity of the items comprising the ACP-PI was verified by a literature review and expert panel using the Delphi method [19]. The factors were established in the EFA for structural validity, and CFA was used to confirm the established factors. The CFA results showed that the RMSEA did not meet the criteria, but the model’s goodness of fit was improved by considering the error correlation from the modified index. The three-factor structure was deemed appropriate because a value <0.08 is considered “adequate” [31], and the other goodness-of-fit indices achieved the required criteria. Internal consistency was confirmed by Cronbach’s alpha coefficients for the overall instrument and for each sub-factor, which were above 0.7 [29]. Thus, the inventory had sufficient reliability and validity. Further verification of reliability and validity, including test–retest reliability, should be conducted in the future.

### 4.2. Description of Factors of the ACP-PI

The factor “Provision of information and conversation with the resident to encourage them to express their end-of-life wishes” identified in this study included items on “providing information about end-of-life,” as well as “discussed end-of-life care”. PWD and their family members may not have the correct knowledge about the end-of-life stage [32]; hence, it is necessary to provide them with this knowledge to avoid unintended consequences. However, in group homes, there are no standard staffing requirements for medical care providers, and the number of staff is lower than in other types of long-term care insurance facilities. Therefore, it is not always possible to provide residents with optimal medical treatment that may be necessary when considering promoting decision making about end-of-life care within the facility [33].

The second factor, “Preparations in case the resident becomes unable to express his/her own end-of-life wish”, includes writing down one’s intentions and determining/informing a surrogate decision maker, like an advance directive. In previous studies, the advance directive has been considered to be a byproduct or part of the ACP [8,34], and our study also confirms this.

The last factor, “Devising to encourage the resident to express his/her wish with consideration for his/her dementia”, requires the inclusion of practices according to the level of cognitive functioning of PWD and provision of appropriate support for PWD to express their intentions. Because staff with less experience in dementia care might have difficulty implementing this practice, the type of training and experience required to facilitate better implementation of this factor should be further investigated.

### 4.3. Association of the ACP-PI with QOD-LTC

This study provides new insights into the effectiveness of ACP by identifying and assessing the ACP-PI’s factor structure and content. The ACP-PI was associated with the QOD-LTC in both the composite score and the score for each factor. Particularly, “Provision of information and conversation with the resident to encourage them to express their end-of-life wish” was more significantly related to the QOD-LTC than with other factors. The QOD-LTC includes items such as practicing end-of-life care that is unique to the individual and maintaining the individual’s dignity [20], and it is presumed that the factor of ACP-PI that encourages the individual to express their intentions directly was significantly related.

There are several limitations to this study. First, the response rate was low, as with previous studies [35], and the responses may have come from facilities that are positively involved in end-of-life care. Second, we asked the care planners to respond to the quality of end-of-life care by recalling a patient who had died in the facility, which is not the same as the quality of end-of-life care as judged by the PWD or their family. Third, while the definition of ACP [8] includes patients, family members, and facility staff, items related to family and staff were eliminated from distribution during the course of the analysis. Finally, since this study only included group homes that provide end-of-life care, it is necessary to examine the applicability of our findings to other similar facilities.

## 5. Conclusions

This study involved developing an inventory to assess the relationship between ACP-PI implementation and residents’ quality of dying in PWD group homes. The inventory allows for a structured view of ACP, unlike the assessment of ACP that has been done in previous studies. The ACP-PI may be applicable to intervention studies involving educational programs to improve the quality of dying and decision support, as well as to studies in settings other than group homes.

## Figures and Tables

**Figure 1 healthcare-10-00062-f001:**
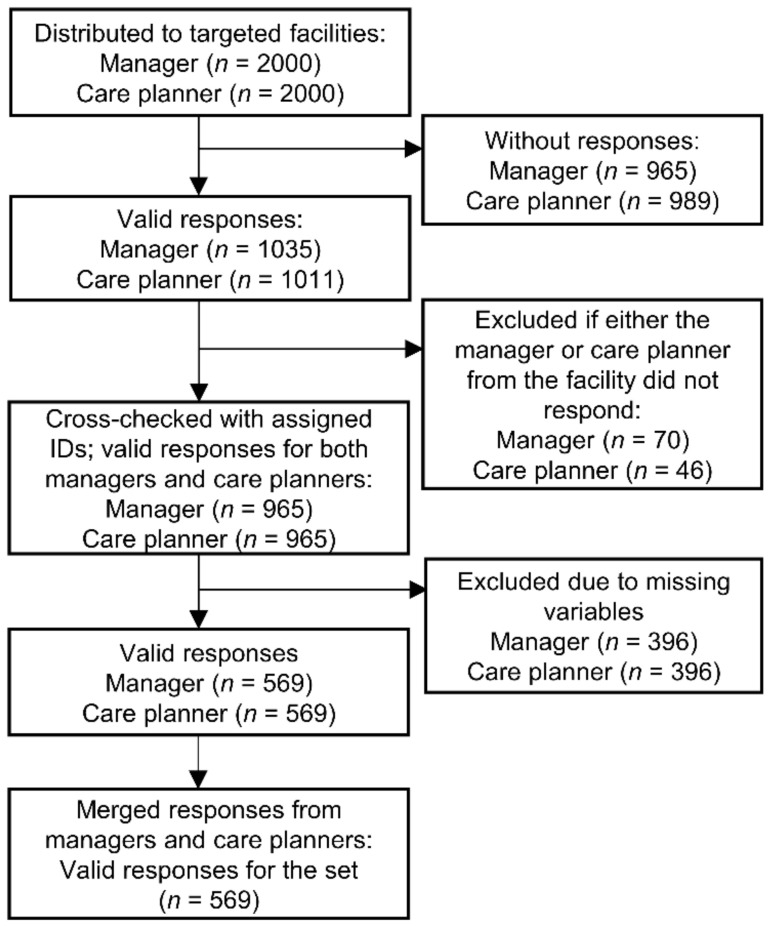
Flow chart of the study participants.

**Table 1 healthcare-10-00062-t001:** Characteristics of the facilities, managers, and care planners (*N* = 569).

Variables	Frequency (%)	Mean (SD)
Facility		
Established by		
Social welfare corporations	115 (20.2)	
Medical corporations	103 (18.1)	
For-profit corporations	300 (52.7)	
Other or no response	51 (9.0)	
Number of units		
1	150 (26.4)	
2	370 (65.0)	
3 or more	38 (6.7)	
No response	11(1.9)	
Year of establishment (A.D.)		2007.0 (6.2)
Number of full-time nurses		0.8 (2.5)
Number of residents		15.9 (6.5)
Managers		
Sex		
Male	229 (40.2)	
Female	333 (58.5)	
No response	7 (1.2)	
Type of work ^†^		
Care worker	429 (75.4)	
Care manager	252 (44.3)	
Nurse	40 (7.0)	
Age (years)		49.8 (10.5)
Experience at the facility (years)		9.8 (5.7)
Experience as a manager (years)		5.9 (4.8)
Care Planners		
Sex		
Male	152 (26.7)	
Female	410 (72.1)	
No response	7 (1.2)	
Type of work ^†^		
Care worker	446 (78.4)	
Care manager	410 (72.1)	
Nurse	32 (5.6)	
Age (years)		50.0 (10.9)
Experience at the facility (years)		8.4 (5.3)
FATCOD-B-J-S score		
Total		23.1 (2.8)
Sub-scale 1		11.4 (1.8)
Sub-scale 2		11.7 (2.0)

^†^ Duplicate answers. Note: SD, Standard deviation; A.D., anno Domini; FATCOD-B-J-S, Japanese version of the Frommelt Attitudes toward Care of the Dying Scale short version. Sub-scale 1: positive attitudes toward caring for dying persons; sub-scale 2: recognition of caring for the pivot dying persons and their families.

**Table 2 healthcare-10-00062-t002:** Characteristics of the recently deceased residents (*N* = 569).

Variables	Frequency (%)	Mean (SD)
Sex		
Male	124 (21.8)	
Female	445 (78.2)	
FAST level at the time of death		6.4 (1.1)
6 or less	224 (39.4)	
7	345 (60.6)	
Presence of end-of-life bonus		
Yes	359 (63.1)	
Length of stay in the facility (years)		4.6 (3.8)
Age at the time of death (years)		91.4 (6.4)
QOD-LTC score		
Composite score		3.3 (0.7)
Sub-scale 1		3.9 (0.6)
Sub-scale 2		3.3 (0.8)
Sub-scale 3		2.7 (1.2)
QOD-LTC score		
High	354 (62.2)	

Note: SD, standard deviation; FAST, functional assessment staging; QOD-LTC, quality of dying in long-term care. QOD-LTC score: individuals with scores ≤3 were categorized as “low group”, and those with scores >3 were categorized as “high group”. Sub-scale 1: personhood; sub-scale 2: preparatory tasks; sub-scale 3: Closure.

**Table 3 healthcare-10-00062-t003:** Items with ceiling effects (*N* = 569).

		Frequency (%)				
No.	Items	Did Not Implement	More or Less Did Not Implement	More or Less Implemented	Implemented	Mean (SD)
23	We asked Mr./Ms. A’s family members and other relevant parties about what Mr./Ms. wished regarding the final stage of life.	16 (2.8)	6 (1.1)	80 (14.1)	467 (82.1)	3.8 (0.6)
24	We talked with A’s family members about Mr./Ms. A’s medical treatment/care options for the final stage of life.	16 (2.8)	2 (0.4)	60 (10.5)	491 (86.3)	3.8 (0.6)
25	We talked with Mr./Ms. A’s family members or other relevant parties about relief and comfort care for distressing symptoms (e.g., breathlessness or discomfort from being unable to move) at the final stage of life.	17 (3.0)	9 (1.6)	72 (12.7)	472 (83.0)	3.8 (0.6)
26	We facilitated and supported Mr./Ms. A’s discussions with their family members so that they could all reach a consensus concerning the policies and procedures regarding the final stage of life.	26 (4.6)	29 (5.1)	112 (19.7)	402 (70.7)	3.6 (0.8)
27	In cases where family members or other relevant parties requested, staff members always accepted calls for consultation regarding the final stage of Mr./Ms. A’s life.	13 (2.3)	4 (0.7)	107 (18.8)	445 (78.2)	3.7 (0.6)
28	After discussing Mr./Ms. A’s medical treatment/care policies with both family and staff members, we documented the details in Mr./Ms. A’s records.	16 (2.8)	16 (2.8)	85 (14.9)	452 (79.4)	3.7 (0.7)
29	We asked family members and relevant parties whether or not there were any changes in their wishes regarding Mr./Ms. A’s medical treatment/care, as needed.	16 (2.8)	22 (3.9)	109 (19.2)	422 (74.2)	3.7 (0.7)
31	We recorded wishes Mr./Ms. A had in relation to the kind of medical treatment/care.	55 (9.7)	53 (9.3)	127 (22.3)	334 (58.7)	3.3 (1.0)
32	Information about Mr./Ms. A, even if it was not medical treatment/care-related, was kept in the records.	7 (1.2)	17 (3.0)	139 (24.4)	406 (71.4)	3.7 (0.6)
33	The medical treatment/care policies for the final stage of Mr./Ms. A’s life were discussed and decided by the team at the facility.	8 (1.4)	22 (3.9)	139 (24.4)	400 (70.3)	3.6 (0.6)
34	We shared Mr./Ms. A’s wishes regarding medical treatment/care with the relevant doctors.	14 (2.5)	11 (1.9)	112 (19.7)	432 (75.9)	3.7 (0.6)
35	We shared Mr./Ms. A’s wishes regarding their medical treatment/care, as well as other relevant matters, with facility staff members.	11 (1.9)	14 (2.5)	128 (22.5)	416 (73.1)	3.7 (0.6)
36	We shared Mr./Ms. A’s wishes about medical treatment/care with the long-term care insurance facilities and medical institution staff with whom we work.	25 (4.4)	34 (6.0)	126 (22.1)	384 (67.5)	3.5 (0.8)
37	We kept a record of observations and significant changes pertaining to Mr./Ms. A.	6 (1.1)	4 (0.7)	85 (14.9)	474 (83.3)	3.8 (0.5)
38	In making decisions about Mr./Ms. A’s medical treatment/care, as well as other matters, we took into consideration the wishes Mr./Ms. A expressed regarding the final stage of life and everyday routines.	41 (7.2)	61 (10.7)	184 (32.3)	283 (49.7)	3.3 (0.9)
39	We took Mr./Ms. A’s quality of life into account when making decisions about medical treatment/care.	4 (0.7)	17 (3.0)	193 (33.9)	355 (62.4)	3.6 (0.6)

Note: SD, standard deviation. Scores range from “1. Did not implement” to “4. Implemented”. The percentages of each item are rounded to the nearest whole number, so the total of the breakdown may not add up to 100%.

**Table 4 healthcare-10-00062-t004:** Final exploratory factor analysis, Cronbach’s α, and descriptive statistics of the ACP-PI (*N* = 228).

No.	Item	Factor Loading		
** *Factor 1. Provision of information and conversation with the resident to encourage them to express their end-of-life wishes* **				
1	We judged from Mr./Ms. A’s words, actions, and appearance whether or not they were willing to talk about the final stage of life.	**0.48**	−0.09	0.15
2	We informed Mr./Ms. A about the medical treatment/care available and how they could spend the final stage of life, as detailed within the facility’s policies.	**0.86**	−0.01	−0.02
3	We informed Mr./Ms. A of the significance of communicating their wishes regarding the final stage of life and everyday routines to family and the facility’s staff members while they were still able to express wishes.	**0.86**	0.06	−0.08
4	We informed Mr./Ms. A of what things they should tell family and staff members while they were still able to express wishes.	**0.77**	0.09	−0.02
5	We presented Mr./Ms. A with specific details about what medical treatment/care options were available as they entered the final stage of life (note: this statement includes cases where part-time doctors or similar staff presented the information).	**0.77**	0.04	−0.01
7	We talked with Mr./Ms. A about what kind of medical treatment/care they wished for at the final stage of life.	**0.93**	−0.01	−0.05
8	We talked with Mr./Ms. A about where they would like to spend the final stages of life.	**0.87**	−0.05	−0.06
9	We discussed with the patient the kind of care they would like to receive to get relief and comfort from distressing symptoms (e.g., breathlessness or discomfort from being unable to move) at the final stage of life.	**0.79**	0.00	0.05
10	We discussed with Mr./Ms. A whether or not there were any changes in the their wishes regarding medical treatment/care each time it happened.	**0.67**	0.02	0.07
11	Whenever Mr./Ms. A’s condition changed, we spoke with them to establish whether or not there were any changes in wishes regarding medical treatment/care.	**0.71**	0.04	0.07
** *Factor 2. Preparations in case the resident becomes unable to express their own end-of-life wish* **				
13	We asked Mr./Ms. A to put their wishes regarding their medical treatment/care, as well as any other wishes, in writing.	0.05	**0.65**	−0.03
14	At Mr./Ms. A’s request, we gave a copy of the written document to their spouse stating their wishes regarding medical treatment/care and other matters.	−0.06	**0.76**	0.01
15	We asked Mr./Ms. A who they wished to participate in discussions about medical treatment/care and other matters relating to the time when they would no longer be able to make their own decisions.	0.04	**0.78**	0.02
16	We asked Mr./Ms. A whether or not they had informed the person they mentioned in Point 15 about their desire to have them participate in such discussions.	0.05	**0.88**	0.00
17	We informed Mr./Ms. A about representatives and/or systems that handle legal aspects after they pass away (e.g., implementing a will).	−0.04	**0.71**	0.02
** *Factor 3. Devising to encourage the resident to express their wish with consideration for their dementia* **				
19	We considered the topics of discussion regarding the final stage of Mr./Ms. A’s life depending on their cognitive functioning.	−0.07	0.02	**0.86**
20	The ways in which I explained the final stage of life were modified depending on Mr./Ms. A’s cognitive functions.	−0.01	−0.01	**0.98**
21	When discussing the final stage of life with Mr./Ms. A, we checked that they understood the content.	0.27	0.03	**0.55**
Cronbach’s α ^☨^		0.94	0.86	0.87
Composite score, mean (SD)		2.0 (0.7)		
Factor 1, mean (SD)		2.1 (0.8)		
Factor 2, mean (SD)		1.5 (0.7)		
Factor 3, mean (SD)		2.5 (0.9)		

Note: ACP, advance care planning; ACP-PI, ACP Practice Inventory; SD, standard deviation. Italics indicate factor names. Exploratory factor analysis: promax rotation, factor loading >0.45. Items with factor loadings of 0.45 or higher and belonging to a factor are expressed in bold. Bartlett’s sphericity test was significant (χ^2^ = 3244.2, df = 153, *p* < 0.01), and the Kaiser–Meyer–Olkin value was 0.92. ^☨^ Cronbach’s α (overall) = 0.94.

**Table 5 healthcare-10-00062-t005:** The final version of Advance Care Planning Practice Inventory (ACP-PI).

Instruction text: Circle the Corresponding Number to Indicate Which of the Following Items Have Been Conducted in Relation to Mr./Ms. A.					
No.	Items	Did not Implement	More or less Did not Implement	More or less Implemented	Implemented
1	We judged from Mr./Ms. A’s words, actions, and appearance whether or not they were willing to talk about the final stage of life.	1	2	3	4
2	We informed Mr./Ms. A about the medical treatment/care available and how they could spend the final stage of life, as detailed within the facility’s policies.	1	2	3	4
3	We informed Mr./Ms. A of the significance of communicating their wishes regarding the final stage of life and the everyday routines to the family and the facility’s staff members while they were still able to express wishes.	1	2	3	4
4	We informed Mr./Ms. A of what things they should tell family and staff members while they were still able to express wishes.	1	2	3	4
5	We presented Mr./Ms. A with specific details about what medical treatment/care options were available as they entered the final stage of life (note: this statement includes cases where part-time doctors or similar staff presented the information).	1	2	3	4
6	We talked with Mr./Ms. A about what kind of medical treatment/care they wished for at the final stage of life.	1	2	3	4
7	We talked with Mr./Ms. A about where they would like to spend the final stages of life.	1	2	3	4
8	We discussed with the patient about the kind of care they would like to receive to get relief and comfort from distressing symptoms (e.g., breathlessness or discomfort from being unable to move) at the final stage of life.	1	2	3	4
9	We discussed with Mr./Ms. A about whether or not there were any changes in the wishes of Mr./Ms. A regarding medical treatment/care, each time it happens.	1	2	3	4
10	Whenever Mr./Ms. A’s condition changed, we spoke with them to establish whether or not there were any changes in wishes regarding medical treatment/care.	1	2	3	4
11	We asked Mr./Ms. A to put their wishes regarding their medical treatment/care, as well as any other wishes, in writing.	1	2	3	4
12	At Mr./Ms. A’s request, we gave a copy of the written document to Mr./Ms. A stating the wishes regarding medical treatment/care and other matters.	1	2	3	4
13	We asked Mr./Ms. A who they wished to participate in discussions about medical treatment/care and other matters relating to the time when they would no longer be able to make their own decisions.	1	2	3	4
14	We asked Mr./Ms. A whether or not they had informed the person they mentioned in Point 17 about their desire to have them participate in such discussions.	1	2	3	4
15	We informed Mr./Ms. A about representatives and/or systems that handle legal aspects after they pass away (e.g., implementing a will).	1	2	3	4
16	We considered the topics of discussion regarding the final stage of Mr./Ms. A’s life depending on their cognitive functioning.	1	2	3	4
17	The ways in which I explained the final stage of life were modified depending on Mr./Ms. A’s cognitive functions.	1	2	3	4
18	When discussing the final stage of life with Mr./Ms. A, we checked that they understood the content.	1	2	3	4

**Table 6 healthcare-10-00062-t006:** Logistic regression for the residents’ quality of dying and related factors (*N* = 569).

	Model 1	Model 2	Model 3	Model 4	Model 5
	OR (95%CI)	OR (95%CI)	OR (95%CI)	OR (95%CI)	OR (95%CI)
Residents’ characteristics					
Female sex ^†^	0.90 (0.57–1.42)	0.97 (0.61–1.54)	0.91 (0.58–1.42)	0.87 (0.55–1.35)	0.93 (0.59–1.48)
Length of stay (years) ^‡^	0.92 (0.54–1.55)	0.87 (0.52–1.48)	0.88 (0.52–1.47)	0.95 (0.56–1.59)	0.89 (0.53–1.51)
Age at death (years)	1.05 (1.02–1.08) **	1.05 (1.02–1.08) **	1.05 (1.02–1.09) **	1.05 (1.02–1.08) **	1.05 (1.02–1.08) **
FAST ^§^	1.10 (0.76–1.61)	1.12 (0.76–1.63)	1.06 (0.73–1.53)	1.03 (0.71–1.48)	1.12 (0.77–1.64)
GHPWD’s characteristics					
Number of full-time nurses	1.13 (1.03–1.25) *	1.13 (1.02–1.24) *	1.12 (1.02–1.24) *	1.13 (1.02–1.25) *	1.13 (1.03–1.25) *
End-of-life care bonus ^¶^	1.64 (1.12–2.38) *	1.69 (1.16–2.47) **	1.57 (1.08–2.27) *	1.67 (1.16–2.42) **	1.66 (1.13–2.42) **
ACP-PI					
Composite score	2.62 (1.95–3.53) ***				
Factor 1		2.15 (1.70–2.73) ***			1.75 (1.28–2.40) ***
Factor 2			2.16 (1.57–2.97) ***		1.27 (0.87–1.87)
Factor 3				1.65 (1.35–2.01) ***	1.17 (0.92–1.49)

Note: OR, odds ratio; 95%CI, 95% confidence interval; FAST, functional assessment staging; GHPWD, group home for persons with dementia; ACP, advance care planning; ACP-PI, ACP Practice Inventory. * *p* < 0.05, ** *p* < 0.01, *** *p* < 0.001. Dependent variable: the QOD-LTC—0 = low, 1 = high. Regarding independent variables, relevant individual and facility attributes were entered in all models; for the ACP-PI, they were entered as follows: model 1: composite score; models 2–4: one for each factor; model 5: all for each factor. ^†^ 0 = male, 1 = female; ^‡^ 0 = less than 1 year, 1 = 1 year or more; ^§^ 0 = FAST level 6 or less, 1 = FAST level 7; ^¶^ 0 = none, 1 = with bonus. Factor 1: provision of information and conversation with the resident to encourage them to express their end-of-life wishes. Factor 2: preparations in case the resident becomes unable to express their own end-of-life wishes. Factor 3: devising to encourage the resident to express their wishes with consideration for their dementia.

## Data Availability

Data sharing is not applicable to this article because we have not obtained approval of the participants or the Ethics Committee to release the data.

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
