# Peer review of "Implementation of an Advance Care Planning Inventory and Its Possible Effect on Quality of Dying: A Nationwide Cross-Sectional Study in Group Homes for Persons with Dementia in Japan"

_healthcare, 2021, doi:10.3390/healthcare10010062_

Round 1

Reviewer 1 Report

This is an interesting cross-sectional study conducted in Japanese to investigate the advance care planning inventory with the quality of dying for persons with dementia using publicly available databases.  The author needs to clarify the points below before publication:1. The exact content of the ACP-PI, please give more detailed information (or as appendix);2. Scoring methodology for ALL included questionnaires;3. The authors mentioned the “multicollinearity” problem without detailed information. Solutions for the problem used to be removing some of the highly correlated independent variables; combining the independent variables, for example adding them together; or performing an analysis designed for highly correlated variables, for example principal components analysis or partial least squares regression. But the authors use “covariates” as a solution (To avoid multicollinearity, variables related to residents and facilities were included as covariates, based on existing studies and empirical data), please clarify;4. In their analysis part, please clarify data distribution; differences between model 1-5?  5. In the logistic regression analysis, for example model 5, because all ACP-PI factors were simultaneously entered into the model, please clarify if multiple testing correction was done to reduce the chance of a type I error.6. Some English grammar issues.

Reviewer 2 Report

This is an impressive work with interesting and intriguing results. The correlation between the implementation of an ACP inventory with quality of dying scales is interesting. It might however be confounded by all sorts of factors: attracting attention to ACP might increase awareness for end of life care, might increase selection bias towards those centres more interested in end of life care, might increase education etc.

I think the title does not really represent the manuscript. Something like: "Implementation of an ACP-inventory and its possible effect on quality of dying: a nationwide cross sectional study in group houses for patients with dementia in Japan." might be better. 

Reviewer 3 Report

  1. What is the training program for the research assistants to ensure the inter-rater reliability? (lines 71-74).
  2. Please list the content validity index of the initial ACP-PI. (lines 92-94).
  3. The authors need to describe the frameworks of ACP-PI, and the examples of items (lines 95-100).
  4. The reasons to add one more item need to be stated (line 96).
  5. The results of pilot survey have to be briefly described to support no revisions of 39 items (lines 96-100).
  6. Please explain the calculation of the valid response rate 28.5% (lines 174-175).
  7. Table 1 and Table 2 need to add the frequency and percentage.
  8. There are 21 items in Table 4 not 22 items.
  9. The data in the Table 4 are not clear, eg. there are three numbers in the factor loading.
  10. Both of the EFA and CFA are the methods of constructive validity. The first step is to conduct EFA and establish the frameworks or domains of the scale (inventory). Then using CFA to validate the established frameworks or domains. But these process was not clear in this manuscript, especially CFA (lines 197-210). CFA needs to conduct many modifications of models.
  11. The results of logistic regression in Table 5 need to be explained more detail and clearly.
  12. “The content validity of the ACP-PI was ensured by a literature review and an expert panel” (lines 240-242). Please point out the evidences to support this discourse.
  13. There are some questions which need to be clarified in the section of result. Hence, the discourse in the section of discussion is difficult to judge.

Round 2

Reviewer 1 Report

No special comments. 

Reviewer 3 Report

"As the recovery rate was low at 13%, it was decided that further 114
surveys would need to confirm the results, and the survey was adopted without any re- 115visions. Thus, ultimately, a 39-item list was developed." What is this meaning?
